# VP2 residue N142 of coxsackievirus A10 is critical for the interaction with KREMEN1 receptor and neutralizing antibodies and the pathogenicity in mice

Xue Li[1,2], Zeyu Liu[1,2], Xingyu Yan[1], Yuan Tian[3], Kexin Liu[1], Yue Zhao[3], Jiang Shao[3], Pei Hao[2], Chao Zhang[1] *

1 Shanghai Institute of Infectious Disease and Biosecurity, Fudan University, Shanghai, China, 2 Shanghai Institute of Immunity and Infection, Chinese Academy of Sciences, University of Chinese Academy of Sciences, Shanghai, China, 3 Institutional Center for Shared Technologies and Facilities of Shanghai Institute of Immunity and Infection, Chinese Academy of Sciences, Shanghai, China

* chao_zhang@fudan.edu.cn

**Data Availability Statement:** All relevant data are provided as figures within the paper.

## Abstract

Coxsackievirus A10 (CVA10) has recently emerged as one of the major causative agents of hand, foot, and mouth disease. CVA10 may also cause a variety of complications. No approved vaccine or drug is currently available for CVA10. The residues of CVA10 critical for viral attachment, infectivity and in vivo pathogenicity have not been identified by experiment. Here, we report the identification of CVA10 residues important for binding to cellular receptor KREMEN1. We identified VP2 N142 as a key receptor-binding residue by screening of CVA10 mutants resistant to neutralization by soluble KREMEN1 protein. The receptor-binding residue N142 is exposed on the canyon rim but highly conserved in all naturally occurring CVA10 strains, which provides a counterexample to the canyon hypothesis. Residue N142 when mutated drastically reduced receptor-binding activity, resulting in decreased viral attachment and infection in cell culture. More importantly, residue N142 when mutated reduced viral replication in limb muscle and spinal cord of infected mice, leading to lower mortality and less severe clinical symptoms. Additionally, residue N142 when mutated could decrease viral binding affinity to anti-CVA10 polyclonal antibodies and a neutralizing monoclonal antibody and render CVA10 resistant to neutralization by the anti-CVA10 antibodies. Overall, our study highlights the essential role of VP2 residue N142 of CVA10 in the interactions with KREMEN1 receptor and neutralizing antibodies and viral virulence in mice, facilitating the understanding of the molecular mechanisms of CVA10 infection and immunity. Our study also provides important information for rational development of antibody-based treatment and vaccines against CVA10 infection.

**Funding:** C.Z. is supported by the Shanghai Municipal Science and Technology Major Project (ZD2021CY001) and Shanghai Rising-Star Program (21QA1410000). The funders had no role in study design, data collection and analysis, decision to publish, or preparation of the manuscript.

**Competing interests:** The authors have declared that no competing interests exist.

## Author summary

Coxsackievirus A10 (CVA10) is an emerging pathogen that causes hand, foot, and mouth disease. No approved vaccines or antiviral treatments for CVA10 infection are yet available. Here we identified VP2 N142 of CVA10 as a key residue essential for viral recognition by KREMEN1 receptor by selection of soluble receptor neutralization-resistant mutant and mutational analysis. Residue N142 when mutated could significantly reduce receptor-binding activity, viral attachment, and infection in vitro. Mouse infection experiments show that residue N142 when mutated could reduce fatality rate and severity of symptoms by decreasing viral loads in limb muscle and spinal cord. Furthermore, residue N142 when mutated could confer resistance to neutralization by anti-CVA10 neutralizing antibodies. Additionally, N142, which is exposed on the canyon rim but highly conserved in CVA10, is identified as a key receptor-binding residue, providing a counterexample to the canyon hypothesis. Overall, our findings promote a better understanding of the molecular basis of CVA10 attachment and infection and in vivo pathogenicity, and also provide useful information for rational research and development of vaccines and antibody-based treatment against CVA10 infection.

## Introduction

Coxsackievirus A10 (CVA10) is a viral pathogen that causes hand, foot and mouth disease (HFMD), which is a common infectious disease and usually affects infants and young children [1]. Moreover, CVA10 may also cause a variety of complications, including cough, bronchitis, pneumonia, diarrhea, vomit, myocarditis, cardiac damage, meningitis, encephalitis, brain myelitis, coma, acute flaccid paralysis, or seizures [2,3]. CVA10 infections are generally mild and self-limiting, but may be severe and life threatening in risk groups such as infants [3,4]. In recent years, numerous HFMD cases and outbreaks associated with CVA10 infection have been reported in China and many other countries worldwide [5–15], highlighting that CVA10 has become one of the major causative agents of HFMD. For now, no approved vaccine or drug is available for CVA10, although several CVA10 vaccine candidates have been tested in animal models [16–21].

CVA10 (prototype strain Kowalik) was first isolated in the United States in 1950 [22]. It belongs to species A of the genus *Enterovirus* in the family *Picornaviridae* [23]. Like other enteroviruses, CVA10 is a non-enveloped, single-stranded RNA virus and has an icosahedral capsid comprised of 60 copies of VP1, VP2, VP3, and internal VP4 [23]. Cryo-electron microscopy (cryo-EM) has been used to study the 3D structures of CVA10 viral particles, including mature virion, A particle (uncoating intermediate), and procapsid (empty capsid). The structures reveal that on the CVA10 virion surface, there are a star-shaped plateau on each of the 5-fold axes, a depression (termed the canyon) encircling the plateau, and a propeller-like protrusion centered on each 3-fold axis [24–28]. The cell surface protein KREMEN1 (KRM1) has been identified as an entry receptor for CVA10 [29]. KRM1 selectively binds to mature CVA10 virion above the canyon [26,28].

Key residues in the CVA10 capsid proteins that are critical for viral binding to KRM1 and viral infection have not been identified by experiment, such as mutational analysis. In this study, we identified VP2 N142 as a key residue responsible for CVA10 interaction with KRM1 receptor by screening of CVA10 mutants that were resistant to neutralization by soluble KRM1 protein. N142 is located at the tip of VP2 EF loop and exposed on the virion surface. We demonstrated that residue N142 when mutated drastically reduced CVA10 attachment

and infection in cell culture by reducing virus binding to KRM1 receptor. More importantly, mouse infection experiments showed that residue N142 when mutated could reduce fatality rate and severity of symptoms by decreasing viral loads in limb muscle and spinal cord, indicating the importance of N142 for the pathogenicity of CVA10 in vivo. Furthermore, residue N142 was shown to be a critical part of the epitopes recognized by anti-CVA10 polyclonal antibodies and a neutralizing monoclonal antibody (MAb) called 2A11. N142 when mutated could confer resistance to neutralization by the anti-CVA10 neutralizing antibodies. Overall, our study demonstrates that VP2 residue N142 of CVA10 is a key residue that plays an important role in the interactions with KRM1 receptor and neutralizing antibodies and viral virulence in mice.

## Results

### Isolation of CVA10 mutants resistant to neutralization with soluble KRM1 receptor

We generated a soluble form of human KRM1 consisting of the ectodomain (residues A23 to G373) fused to the Fc portion of human IgG1, designated KRM1-Fc. The KRM1-Fc protein was able to effectively neutralize CVA10 prototype strain Kowalik with neutralization concentration (100% protection) of 12.5 nM (Fig 1A and 1B). To identify the CVA10 residues involved in KRM1 receptor binding, we attempted to isolate and characterize CVA10 mutants that escaped neutralization with KRM1-Fc (Fig 1A and 1B). CVA10/Kowalik was subjected to three passages in the presence of gradually increasing concentrations of KRM1-Fc (Fig 1A). The neutralization-resistant mutants readily emerged when treated with the appropriate concentrations of KRM1-Fc (Fig 1B). The mutants from two different wells (the upper and lower wells were designated well #1 and #2, respectively) of the culture plate were subjected to plaque purification, and the sequences of the capsid protein-coding region of these isolates were determined and aligned with that of wild-type CVA10/Kowalik. The information of the plaque-purified isolates is summarized in Fig 1B. Among the 10 isolates from well #1, 7 had double mutations: N→D at residue 142 of VP2 (N2142D) and N→Y at residue 183 of VP3 (N3183Y), and 2 had an additional mutation at position 1287 (A1287T) or 2156 (T2156S) besides the N2142D+N3183Y mutation, and the last one possessed a double mutation in the VP1 protein (T1219I and S1238G). Note that viral capsid residues are numbered from 1001, 2001, and 3001 in VP1, VP2, and VP3, respectively. Among the 9 isolates from well #2, 4 had the N3183Y substitution, and the other 4 possessed double mutations at residues 1219 and 1238 (T1219I and S1238G), and the remaining one had a double mutation (N2142D+N3183Y). Thus, there are three major types of mutations: N2142D+N3183Y, N3183Y, and T1219I+S1238G (Fig 1C), which were used for the subsequent analyses.

### The mutation at residue N2142 is responsible for the soluble receptor-resistant phenotype

Each of the mutants N2142D+N3183Y, N3183Y, and T1219I+S1238G was titrated by $TCID_{50}$ assay and then tested for resistance to neutralization by KRM1-Fc by standard neutralization assay. The results are summarized in Fig 2A. The titers of the mutants N3183Y and T1219I +S1238G were comparable to that of wildtype CVA10/Kowalik (CVA10-WT), while the titer of the mutant N2142D+N3183Y were 25 times lower than that of CVA10-WT, indicating that N2142D, but not the other mutations, may reduce viral fitness. Furthermore, the mutants containing the N3183Y substitution or the T1219I+S1238G mutation were sensitive to neutralization with KRM1-Fc, whereas the mutant N2142D+N3183Y was resistant to KRM1-Fc and the

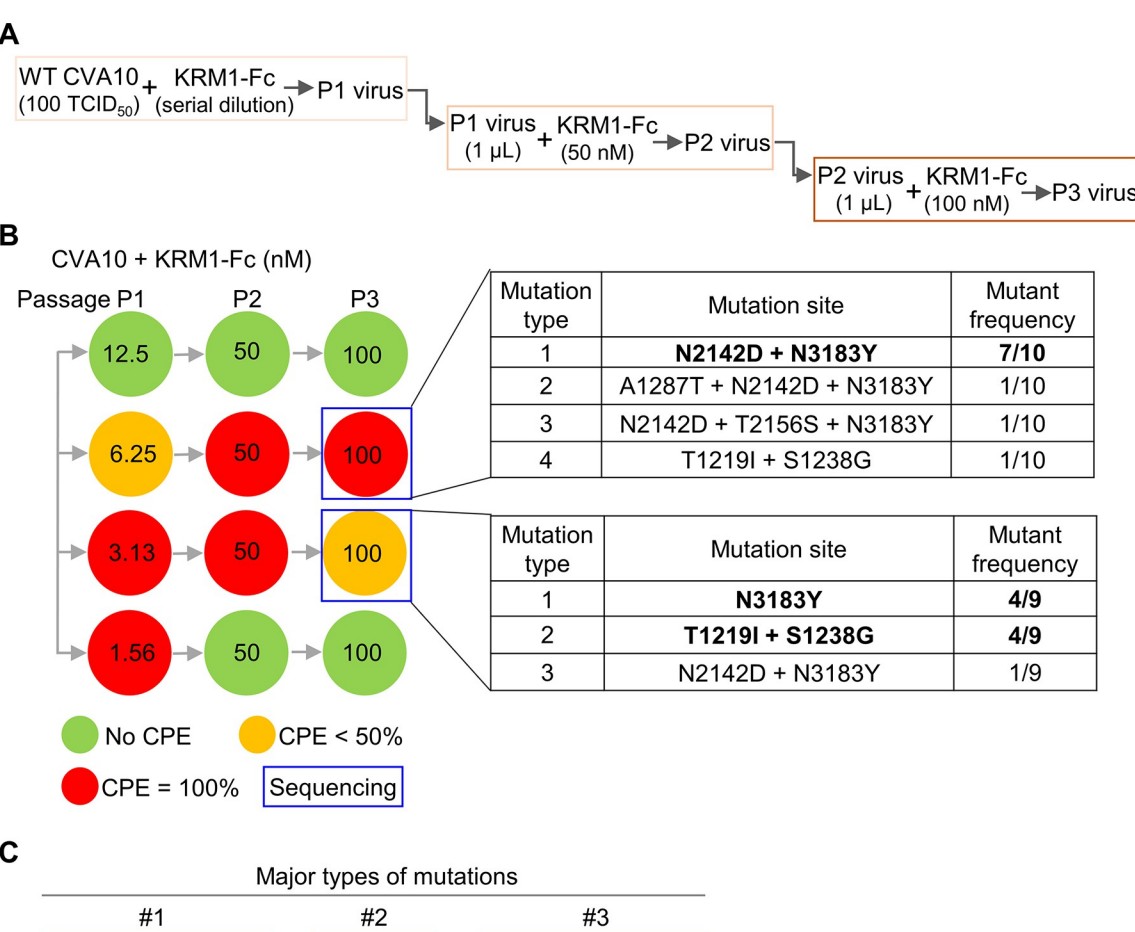

**Fig 1. Selection of CVA10 mutants resistant to neutralization with soluble human KRM1 receptor.** (**A**) A flowchart of the screening procedure. (**B**) Screening and information of soluble KRM1-resistant mutants. Green circle, no CPE; orange, partial CPE; red, complete CPE. 10 and 9 plaques were isolated from well #1 (the upper well) and #2 (the lower well), respectively, and then sequenced to identify mutations. Viral capsid residues are numbered from 1001, 2001, and 3001 in VP1, VP2, and VP3, respectively. (**C**) A summary of the three major types of the mutations. The red arrows indicate the substitutions in the first or second nucleotides of the codons.

neutralization concentration decreased by more than 128 times compared with CVA10-WT. These results suggest that N2142D, but not the N3183Y, T1219I, and S1238G mutations, is likely to be responsible for the soluble receptor-resistant phenotype.

To further confirm the above conclusion, a T7 promoter-driven infectious clone of CVA10/Kowalik was constructed, and the mutations, N3183Y, N3183A, N2142D, T1219I, and S1238G, were separately introduced into the CVA10 infectious clone using site-directed muta-genesis (Fig 2B). The N3183A mutation was designed to change the polarity of N3183 residue from polar to non-polar. The wildtype and mutant CVA10 viruses were rescued by co-trans-fecting HEK 293T cells with CVA10 infectious clone plasmid and the T7 RNA polymerase-encoding plasmid and amplified in RD cells (Fig 2C). The rescued CVA10 viruses were

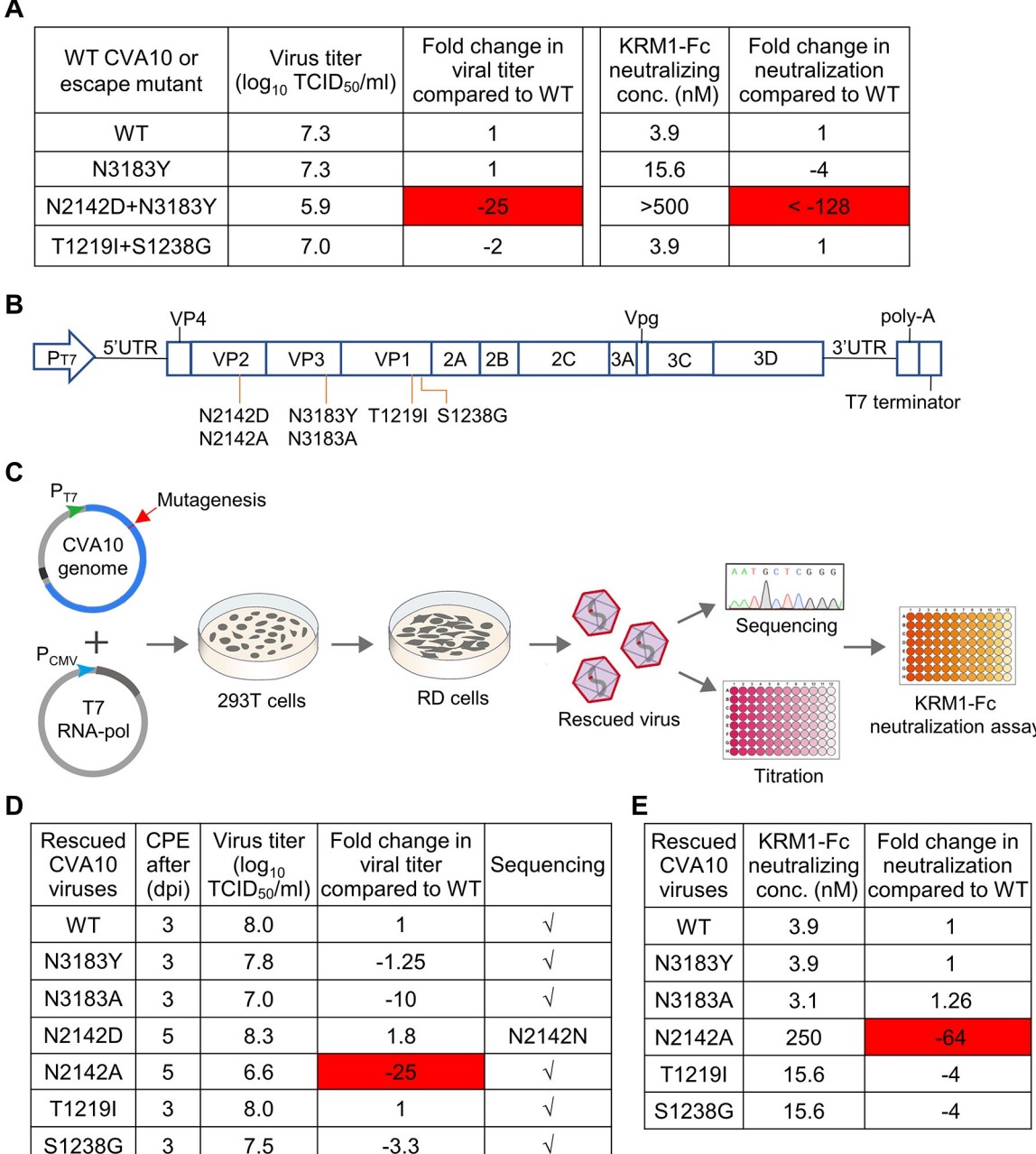

**A**

| WT CVA10 or escape mutant | Virus titer ($\log_{10}$ $TCID_{50}$/ml) | Fold change in viral titer compared to WT | KRM1-Fc neutralizing conc. (nM) | Fold change in neutralization compared to WT |
|---|---|---|---|---|
| WT | 7.3 | 1 | 3.9 | 1 |
| N3183Y | 7.3 | 1 | 15.6 | -4 |
| N2142D+N3183Y | 5.9 | -25 | >500 | < -128 |
| T1219I+S1238G | 7.0 | -2 | 3.9 | 1 |

**D**

| Rescued CVA10 viruses | CPE after (dpi) | Virus titer ($\log_{10}$ $TCID_{50}$/ml) | Fold change in viral titer compared to WT | Sequencing |
|---|---|---|---|---|
| WT | 3 | 8.0 | 1 | √ |
| N3183Y | 3 | 7.8 | -1.25 | √ |
| N3183A | 3 | 7.0 | -10 | √ |
| N2142D | 5 | 8.3 | 1.8 | N2142N |
| N2142A | 5 | 6.6 | -25 | √ |
| T1219I | 3 | 8.0 | 1 | √ |
| S1238G | 3 | 7.5 | -3.3 | √ |

**E**

| Rescued CVA10 viruses | KRM1-Fc neutralizing conc. (nM) | Fold change in neutralization compared to WT |
|---|---|---|
| WT | 3.9 | 1 |
| N3183Y | 3.9 | 1 |
| N3183A | 3.1 | 1.26 |
| N2142A | 250 | -64 |
| T1219I | 15.6 | -4 |
| S1238G | 15.6 | -4 |

**Fig 2. The role of the mutations in conferring resistance to neutralization with soluble KRM1-Fc. (A)** Wildtype and mutant CVA10 viruses were titrated by $TCID_{50}$ assay and analyzed for resistance to neutralization with KRM1-Fc. Fold increase or decrease (prefixed with a minus sign) in viral titer and neutralization concentration of KRM1-Fc against the mutants, relative to wildtype CVA10. Red highlighting, more than 10-fold decrease. **(B)** Schematic representation of construction of the T7 promoter ($P_{T7}$)-driven infectious clone of CVA10. The indicated mutations were separately introduced into the infectious clone. **(C)** Schematic of rescue and characterization of wildtype and mutant CVA10 viruses. T7 RNA-pol, T7 RNA polymerase. **(D)** The rescued CVA10 viruses were titrated by $TCID_{50}$ assay and sequenced. dpi, day post infection. **(E)** The rescued CVA10 viruses were analyzed for resistance to neutralization with KRM1-Fc.

sequenced and titrated by $TCID_{50}$ assay. The results are summarized in Fig 2D. CVA10 viruses WT, N3183Y, N3183A, T1219I, and S1238G were successfully rescued and their infectious viral titers were greater than or equal to $10^7$ $TCID_{50}$/ml. We failed to rescue the N2142D

mutant, because reverse mutation (D→N at residue 2142) occurred in cell culture. To address this issue, we constructed a new CVA10 mutant N2142A and found that this mutant could be successfully rescued using the reverse genetics and was genetically stable, although its infectious viral titer was significantly lower than that of the rescued CVA10-WT (Fig 2B–2D). Next, each of the rescued viruses, WT, N3183Y, N3183A, N2142A, T1219I, and S1238G, was tested for resistance to neutralization by KRM1-Fc. Compared with the rescued CVA10-WT, rescued viruses N3183Y, N3183A, T1219I, and S1238G were still sensitive to neutralization with KRM1-Fc, while N2142A showed 64-fold increase of resistance to KRM1-Fc (Fig 2E). Together, these results demonstrate that the mutation at residue N2142, but not N3183, T1219, and S1238, is responsible for the soluble receptor-resistant phenotype.

## Location and conservation of the mutated residues

The structure of one asymmetric unit of CVA10 in complex with KRM1 receptor (PDB: 7BZU) [28] is shown in Fig 3A. KRM1 receptor binds to the canyon and interacts with the north and south rims and the canyon floor. Residue N2142 is located in the VP2 EF loop and at the south rim of the canyon and makes contact with residue D90 of KRM1 (Fig 3A and 3B). Moreover, the ND2 atom of N2142 forms hydrogen bonds with the backbone oxygen groups of residues D88 and G89 of KRM1, as revealed by PISA analysis (Fig 3C). The impact of mutation of N2142 on the CVA10-KRM1 interaction was analyzed by using the SWISS-MODEL modeling server [30]. The hydrogen bond interactions with D88 and G89 of KRM1 would be lost when N2142 is mutated to Ala or Asp (Fig 3C). Furthermore, the N2142D mutation may result in strong electrostatic repulsive interactions with the negatively charged carboxyl groups of D88 and D90 in KRM1 (Fig 3C). The above structural analysis explains the observed resistance of the N2142A and N2142D mutants to neutralization by KRM1-Fc (Fig 2A and 2E).

Residue N3183 is situated in the VP3 GH loop and on the floor of the canyon and does not form any hydrogen bonds or salt bridge interactions with KRM1 (Fig 3A and 3B), implying that N3183 does not play an important role in the CVA10-KRM1 interaction. Residue T1219 is located in the VP1 GH loop and midway up the south canyon wall and forms a hydrogen bond with H194 of KRM1 (Fig 3A and 3B). Residue S1238 is located near the 5-fold symmetry axis of the virus and is far away from the KRM1 receptor–binding site (Fig 3A), suggesting that S1238 is unlikely to be involved in interactions with KRM1. Therefore, only the N2142, N3183, and T1219 residues were subjected to the subsequent sequence conservation analysis.

A total of 255 CVA10 VP2 protein sequences were downloaded from NCBI database (as of April 2023) using BLAST and aligned using Vector NTI. The analysis showed that 253 VP2 sequences had an asparagine (N) at position 2142, whereas only one VP2 sequence had an aspartic acid (D) at this position (Fig 3D), indicating that CVA10-N2142 has a better viral fitness than CVA10-D2142. In addition, residue N3183 was completely conserved in all of 254 CVA10 VP3 protein sequences downloaded from NCBI (Fig 3D), probably because N3183 is located on the canyon floor and is sequestered away from immune pressures. Among 867 CVA10 VP1 protein sequences from NCBI, 854 had a threonine (T) at position 1219, and 12 had an isoleucine (I) at this position (Fig 3D), indicating that the T1219I mutation occurs in nature.

## The N2142A mutation could reduce viral replication and binding to cell by impairing receptor binding

Growth kinetics of rescued CVA10-WT and mutant N2142A (CVA10-N142A) were compared. RD cells were infected at a multiplicity of infection (MOI) of 0.01 and viral titers were determined at 0, 6, 12, 24, 36, and 48 hours post infection by $TCID_{50}$ assay (Fig 4A). Both

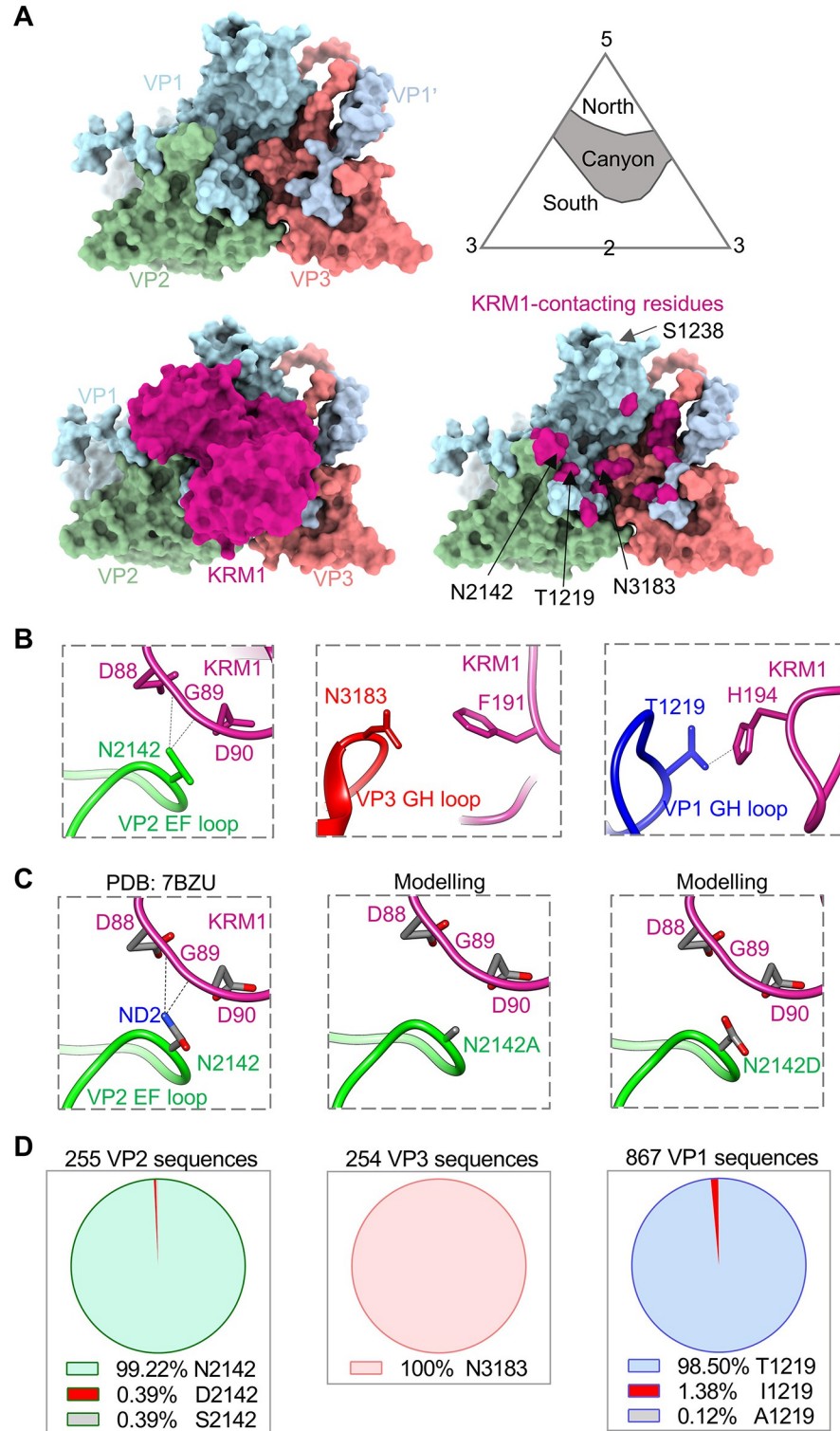

**Fig 3. Location and conservation of the mutated residues.** (**A**) Surface representation of the structure of CVA10 alone and in complex with KRM1 receptor (PDB: 7BZU). Upper left panel: the structure of one asymmetric unit of CVA10. VP1, light blue; VP2, dark sea green; VP3, light coral. VP1', the C terminus of an adjacent VP1 is colored in light steel blue. Upper right panel: the canyon topology. 5, 3, and 2: five-, three-, and two-fold axis. Lower left panel: the structure of one asymmetric unit of CVA10 in complex with KRM1. KRM1, medium violet red. Lower right panel: KRM1-contacting residues of CVA10 (distance cut-off: 4 Å) are colored in medium violet red. (**B**) The detailed

interactions between viral residues (N2142, N3183, and T1219) and surrounding KRM1 residues. Black dash lines indicate hydrogen bonds. **(C)** The impact of mutation of N2142 on the virus-receptor interaction was analyzed using SWISS-MODEL. Nitrogen atom, blue; oxygen, red; carbon, gray. **(D)** Sequence conservation analysis of CVA10 residues 2142, 3183, and 1219. CVA10 VP2, VP3, VP1 protein sequences were obtained from NCBI database and used for this analysis.

CVA10-WT and CVA10-N142A viruses replicated well in RD cells. There were no significant differences in viral titers at 0, 6, and 12 hours post infection between the two groups. However, CVA10-N142A grew significantly worse than CVA10-WT at 24, 36, and 48 hours post infection, and the N2142A mutation resulted in nearly a 10-fold reduction in virus titer, indicating that N2142A mutation is able to significantly reduce CVA10 replication in cells.

To determine the influence of the N2142A mutation on the assembly of CVA10 viral particles, the supernatants of CVA10-WT or CVA10-N142A infected cell cultures were clarified, concentrated, and then subjected to sucrose gradient ultracentrifugation (Fig 4B). SDS-PAGE analysis of the resultant gradient fractions showed that the patterns of capsid protein composition and content were generally similar for the CVA10-WT and CVA10-N142A samples. Both viruses consisted of VP1, VP3, VP0 or/and VP2 proteins, and these viral proteins co-sedimented mainly in fractions #7 to #11. Note that infectious mature CVA10 virion consisted of VP1, VP3, VP2, and VP4 proteins, while non-infectious CVA10 empty particles (also termed procapsid) were composed of VP1, VP3, and VP0 proteins [25]. These results indicate that the surface-exposed N2142A mutation does not affect CVA10 viral assembly. The fraction #10 samples of CVA10-WT and CVA10-N142A were selected for subsequent analysis, because viral particles in this fraction were mainly composed of VP1, VP2, and VP3 proteins, corresponding to mature CVA10 virion.

To verify formation of viral particles, the fraction #10 samples of CVA10-WT and CVA10-N142A were diluted to 50 μg/ml in PBS and subjected to negative stain electron microscopy (EM). As shown in Fig 4C, ~30 nm spherical particles were observed for both virus samples. And importantly, the vast majority of the CVA10-WT and CVA10-N142A virus particles were full particles, corresponding to mature virion. However, CVA10-N142A at the same mass concentration had lower viral titers compared to CVA10-WT. Specifically, viral titers of fraction #10 of CVA10-WT and CVA10-N142A at 1 μg/ml were determined to be $10^{8.5}$ and $10^{6.6}$ TCID$_{50}$/ml, respectively, which was an 86-fold difference (Fig 4D). These results suggest that at the same infectious viral titer, CVA10-N142A has much higher particle number than CVA10-WT.

Viral particles with the same mass concentration, but not the same infectious viral titer, were used to compare the differences between CVA10-WT and CVA10-N142A in terms of cell-binding capacity. 100 ng/ml of CVA10 viral particles (fraction #10) were adsorbed to cooled RD cells at 4°C for 1 h to allow cell binding but not entry, and after washing to remove unbound virus, levels of RNA of CVA10 particles bound to cells were determined by RT-qPCR (Fig 4E). Compared with CVA10-WT, cell-binding activity of CVA10-N142A decreased by 36%, and significant differences were observed between the two groups, indicating that the N2142A mutation could impair virus binding to the cell surface.

We then compared CVA10-WT and CVA10-N142A for binding with soluble human KRM1-Fc protein by ELISA assay. In this assay, CVA10 viral particles (fraction #10) were serially diluted and coated onto ELISA plates, followed by incubation with biotin-labeled human KRM1-Fc protein and HRP-conjugated streptavidin (Fig 4F). CVA10-WT efficiently reacted with human KRM1-Fc in virus dose-dependent fashion, whereas CVA10-N142A reacted with KRM1-Fc only at high virus concentration ($\geq$ 2 μg/ml) but not at low concentrations. Moreover, the KRM1-Fc-binding levels of CVA10-N142A were much lower than those of

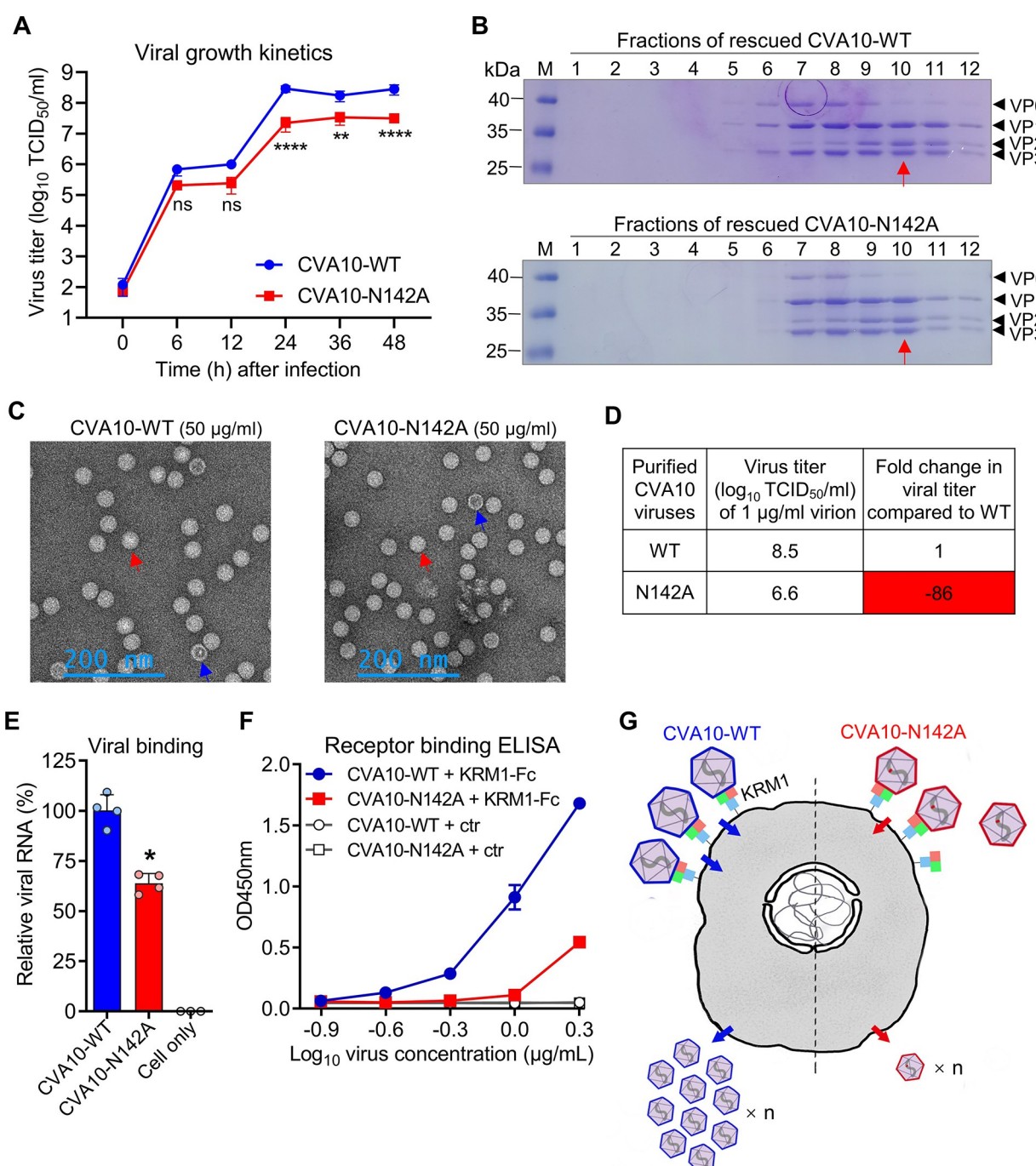

**Fig 4. N2142A mutation reduced viral replication and binding to cell by impairing receptor binding.** (A) Growth kinetics of rescued CVA10-WT and CVA10-N142A were compared in RD cells. RD cells were infected at a MOI of 0.01 and viral titers were determined at the indicated time points post infection. Data are means ± SD of three replicate samples. Analysis was performed using two-way ANOVA. ns, no significant difference ($p \geq 0.05$); **, $p < 0.01$; ****, $p < 0.0001$. (B) CVA10-WT or CVA10-N142A infected cell cultures were subjected to sucrose gradient ultracentrifugation, and the resulting 12 fractions were analyzed by SDS-PAGE. The fraction #10 samples (indicated by red arrows) were selected for subsequent analysis. (C) The fraction #10 samples of CVA10-WT and CVA10-N142A were diluted to 50 µg/ml and subjected to negative stain electron microscopy. Red and blue arrows indicate full (mature virion) and empty CVA10 particles, respectively. (D) The fraction #10 samples of CVA10-WT and CVA10-N142A were diluted to 1 µg/ml and assayed for viral titers. Fold decrease (prefixed with a minus sign) in viral titer is also shown. (E) 100 ng/ml of CVA10 viral particles (fraction #10) were attached to prechilled RD cells at 4°C for 1 h, and after washing, the levels of CVA10 RNA were determined by RT-qPCR. Data are means ± SD of four replicate samples. The Mann–Whitney test was used to compare differences. *, $p < 0.05$. (F) Reactivity of CVA10-WT and CVA10-N142A viral particles with human KRM1-Fc protein were determined by ELISA. CVA10 viral particles were serially diluted and coated onto microplates, followed by incubation with biotinylated

KRM1-Fc or ACE2-Fc (ctr) protein and HRP-conjugated streptavidin. Data are means ± SD of three replicate samples. **(G)** Proposed model of residue N2142-dependent CVA10 attachment and infection of target cells. The N2142A mutation is indicated by red dots.

CVA10-WT. The results demonstrate that the N2142A mutation could reduce CVA10 binding affinity to KRM1 receptor, in line with the results of the cell-attachment assay (Fig 4E).

We proposed a CVA10 infection model to show the importance of residue N2142 in CVA10 attachment and infection processes (Fig 4G). Compared with CVA10-WT, the N2142A mutation could impair virus binding to KRM1 receptor on the cell surface, leading to a significant decrease in the production of progeny virus.

## The N2142A mutation could reduce viral pathogenicity in mice

To evaluate the virulence and pathogenicity of the CVA10 viruses in vivo, groups of 2-day-old ICR mice were infected with 300 $TCID_{50}$ of rescued CVA10-WT or CVA10-N142A, and then clinical symptoms and mortality were observed daily for 14 days. As shown in Fig 5A, CVA10-WT-infected mice rapidly developed limb weakness and paralysis and had mortality rate of 92.3% (12/13). By contrast, only four out of 12 (33.3%) CVA10-N142A-inoculated mice developed limb paralysis and eventually died, and the other eight only had mild symptoms or no symptoms. There was a significant difference in the mortality rate between the two groups (p = 0.0001). The results demonstrate that the N2142A mutation can decrease the virulence of CVA10 in mice.

To determine viral loads in organs of infected mice, groups of ICR mice inoculated with CVA10-WT or CVA10-N142A were sacrificed at 4 days post infection (dpi), and limb muscle, spinal cord, and brain were harvested for determination of virus titers by $TCID_{50}$ assay (Fig 5B). Viral titers per gram of limb muscle tissues from the CVA10-WT and CVA10-N142A groups ranged from 3088 to 902,857 (geometric mean titer = 109,903) and from 4216 to 200,000 (geometric mean titer = 9768), respectively. In addition, viral titers per gram of spinal cord samples from the CVA10-WT and CVA10-N142A groups ranged from undetectable to 920,000 (geometric mean titer = 10,783) and from undetectable to 2107 (geometric mean titer = 290), respectively. There were significant differences in viral titers in limb muscle and spinal cord between the CVA10-WT and CVA10-N142A groups. Moreover, there were significant positive correlations between the viral titers in limb muscle and spinal cord and the clinical scores of infected mice (Fig 5C). However, no statistically significant differences were observed in brain viral titers between the CVA10-WT and CVA10-N142A groups (Fig 5B). Furthermore, no significant correlations between brain viral titers and clinical scores were found (Fig 5C). Taken together, the results imply that the N2142A mutation could reduce the fatality rate and severity of symptoms by decreasing viral loads in limb muscle and spinal cord but not brain.

To determine histopathological changes after infection, groups of ICR mice inoculated with CVA10-WT or CVA10-N142A were sacrificed at 4 dpi, and limb muscle samples were collected and subsequently used for histopathologic examination. Hematoxylin and eosin (H&E) staining images of infected and normal limb muscle tissues are shown in Fig 6A. Note that severity of myositis was graded on a scale of 0 to 4. For the CVA10-WT group, only one out of 15 (6.7%) infected mice showed normal muscle fibers, while the other 14 mice displayed varying degrees of limb muscle tissue damage. 5 out of 11 (45.5%) CVA10-N142A-infected mice showed normal limb muscle morphology, and the other 6 mice developed necrotizing myositis. The CVA10-WT group had a significantly higher mean histopathologic score compared to the CVA10-N142A group (p = 0.0046) (Fig 6B). Therefore, CVA10-N142A-infected mice showed a milder limb muscle pathology compared to CVA10-WT-infected mice. In addition,

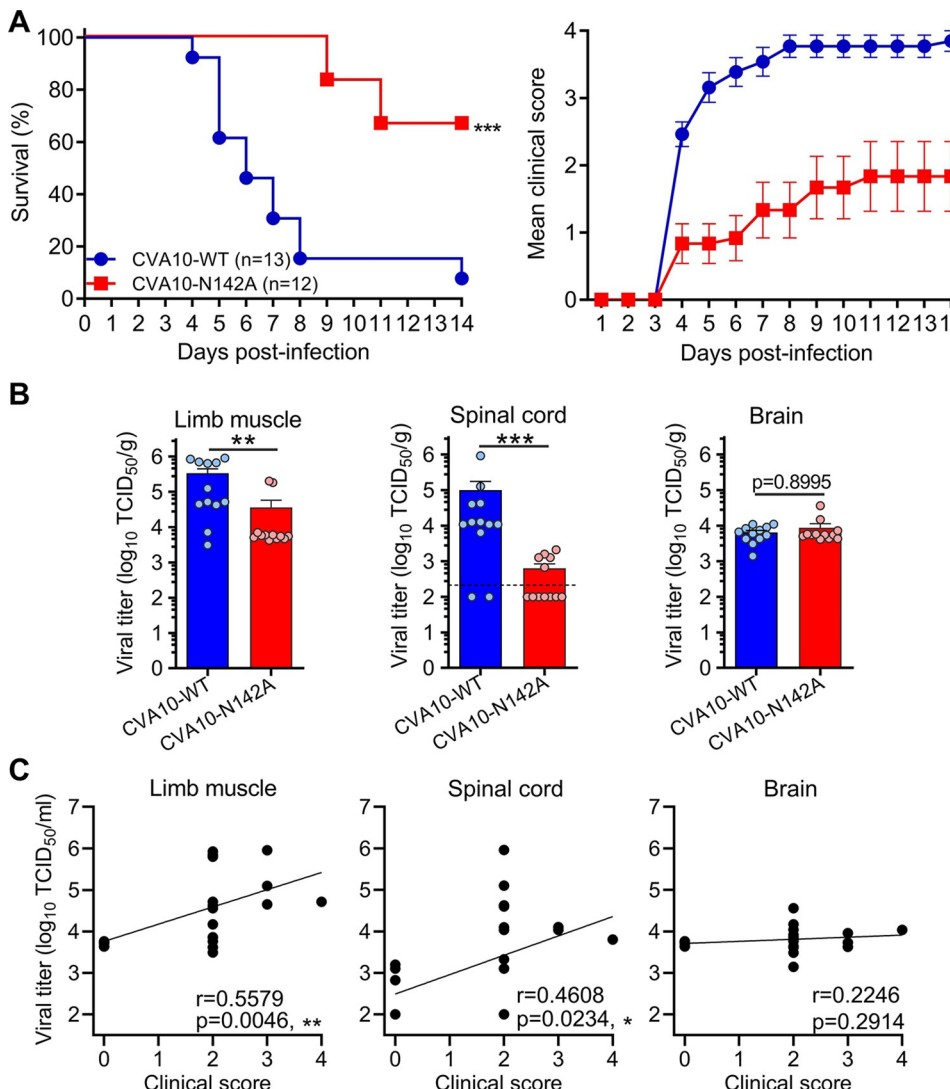

**Fig 5. N2142A mutation could reduce mortality rate and severity of symptoms by decreasing viral loads in limb muscle and spinal cord. (A)** Groups of 2-day-old ICR mice were inoculated with 300 $TCID_{50}$ of CVA10-WT or CVA10-N142A, and then survival (left panel) and clinical symptoms (right panel) were observed daily for 14 days. Clinical scores were graded as follows: 0, healthy; 1, reduced mobility; 2, limb weakness; 3, limb paralysis; 4, death. Number of mice per group were indicated in brackets. Statistical significance of survival curves between groups was determined by Log-rank (Mantel-Cox) test. ***, $p < 0.001$. All error bars represent SEM. **(B)** Groups of ICR mice infected with CVA10-WT or CVA10-N142A were sacrificed at 4 dpi, and limb muscle (left panel), spinal cord (middle panel), and brain (right panel) were collected, weighed, and tested for virus titers. There were 12 mice per group. Each symbol represents an individual mouse. The Mann–Whitney test was used to compare differences. **, $p < 0.01$; ***, $p < 0.001$. **(C)** Correlations between viral titers in the indicated organs and clinical scores. Viral titers and clinical scores of individual infected mice from the CVA10-WT and CVA10-N142A groups were used for Pearson correlation coefficient analysis. *, $p < 0.05$; **, $p < 0.01$.

clinical scores were found to correlate well with the degree of limb muscle tissue destruction (p = 0.0178) (Fig 6C).

To explore why the N142A mutation leads to significantly lower virulence in mice, CVA10-WT and CVA10-N142A were compared for binding with soluble murine KRM1-Fc (mKrm1-Fc) protein by ELISA. mKrm1 was reported to be required for CVA10 infection in mouse, as mKrm1-deficient mice are resistant to CV-A10-induced lethal paralysis [29].

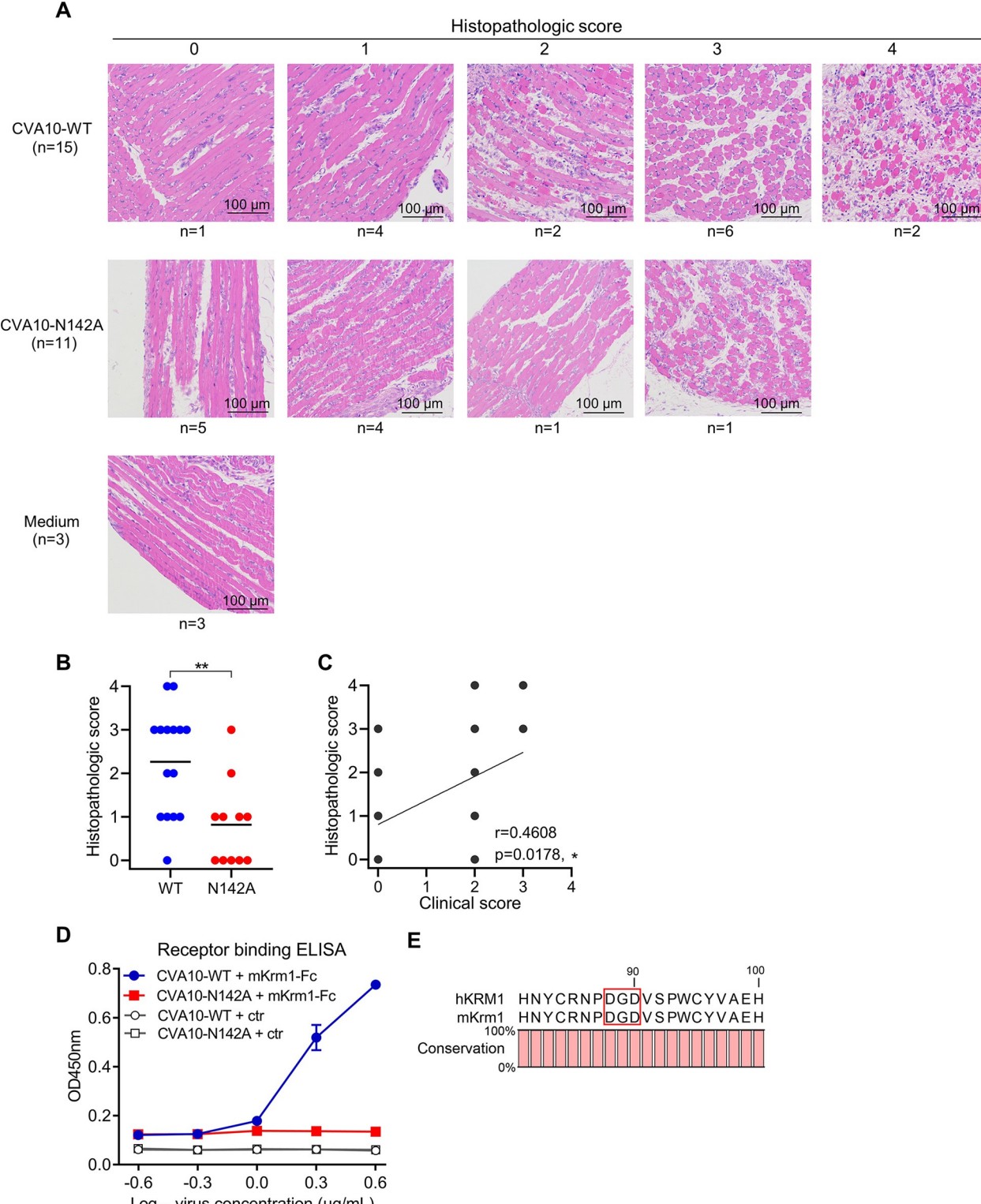

**Fig 6. N2142A mutation could lead to milder limb muscle pathology in infected mice and significantly reduce CVA10 binding affinity to murine KRM1 receptor. (A)** Groups of ICR mice inoculated with medium (control), CVA10-WT, or CVA10-N142A were sacrificed at 4 dpi, and limb muscle samples were collected for histological examination. Histopathologic scores were graded according to the severity of myositis as: not present, mild, moderate, and severe. Number of mice (n) is shown. **(B)** Histopathologic scores of the CVA10-WT and CVA10-N142A groups. Each symbol represents an individual mouse. The Mann–Whitney test was used to compare differences. **, $p < 0.01$. The solid line indicates the mean

values. **(C)** Correlations between histopathologic scores and clinical scores. Histopathologic scores of individual mice from the CVA10-WT and CVA10-N142A groups were used for Pearson correlation analysis. *, $p < 0.05$. **(D)** Reactivity of CVA10-WT and CVA10-N142A viral particles with murine KRM1-Fc (mKrm1-Fc) were analyzed by ELISA. Serially diluted Viral particles were coated onto microplates and then incubated with biotinylated KRM1-Fc or ACE2-Fc (ctr) protein and HRP-conjugated streptavidin. Data are means ± SD of three replicate samples. **(E)** Sequence alignment of the human (hKRM1) and mouse (mKrm1) KRM1 proteins. The three KRM1 residues (D88, G89, and D90) predicted to contact CVA10 residue N142 are boxed. The graph was generated using CLC Sequence Viewer software.

Serially diluted CVA10 viral particles were coated onto ELISA plates and then allowed to incubate with biotin-labeled mKrm1-Fc protein and HRP-conjugated streptavidin (Fig 6D). CVA10-WT could effectively react with mKrm1-Fc in virus dose-dependent manner, whereas CVA10-N142A did not show any obvious binding even at the highest virus concentration tested (4 μg/ml). Thus, the N2142A mutation could significantly reduce CVA10 binding affinity to murine KRM1 receptor. The mouse and human KRM1 proteins share 93.7% sequence identity, and the three residues (D88, G89, and D90) predicted to contact CVA10 residue N142 are identical in the two KRM1 proteins (Fig 6E). When N2142 of CVA10 is mutated to Ala, the hydrogen bond interactions with D88 and G89 of murine KRM1 would be lost, as analyzed above (Fig 3C).

## The N2142A mutation could confer resistance to anti-CVA10 sera and MAb

CVA10-N142A was compared with CVA10-WT by ELISA for the reactivity with polyclonal mouse sera raised against purified CVA10/Kowalik and an anti-CVA10 neutralizing MAb called 2A11. As shown in Fig 7A, both viruses efficiently reacted with anti-CVA10 sera in a virus dose-dependent manner. However, CVA10-N142A showed notably lower sera-binding activity compared to CVA10-WT. In addition, MAb 2A11 bound efficiently to CVA10-WT, but failed to react with CVA10-N142A (Fig 7B). These results indicate that the N142A mutation leads to decreased binding affinities to anti-CVA10 polyclonal antibodies and MAb 2A11, and also suggest that residue N142 is a critical part of the epitopes recognized by the anti-CVA10 antibodies.

Neutralization sensitivity of CVA10-WT and CVA10-N142A viruses to anti-CVA10 polyclonal antibodies was assessed by cell viability assay (Fig 7C). Both viruses were neutralized by anti-CVA10 sera in a titratable manner. The highest serum dilutions that completely prevented cytopathic effect (CPE) induced by CVA10-WT or CVA10-N142A were 64 and 32, respectively. 50% neutralization titers ($NT_{50}$) against CVA10-WT and CVA10-N142A were determined to be 455 and 141, respectively. The N2142A mutation conferred 3.2-fold resistance to neutralization by anti-CVA10 sera. In addition, we tested the neutralization sensitivity of CVA10-WT and CVA10-N142A to MAb 2A11 (Fig 7D). CVA10-WT is highly sensitive to neutralization with MAb 2A11, while CVA10-N142A was highly resistant against 2A11. Specifically, half-maximal inhibitory concentration (IC50) value of MAb 2A11 against CVA10-N142A was determined to be >10 μg/mL, in sharp contrast to that (0.1 μg/mL) against CVA10-WT. These results indicated that N2142A mutation could confer resistance to neutralization by anti-CVA10 neutralizing antibodies.

## Discussion

In this study, we identified N2142 as a key receptor-binding residue by screening of CVA10 mutants resistant to neutralization with soluble KRM1 protein. Residue N2142 of CVA10 plays an important role not only in viral attachment and infection in cell culture, but also in

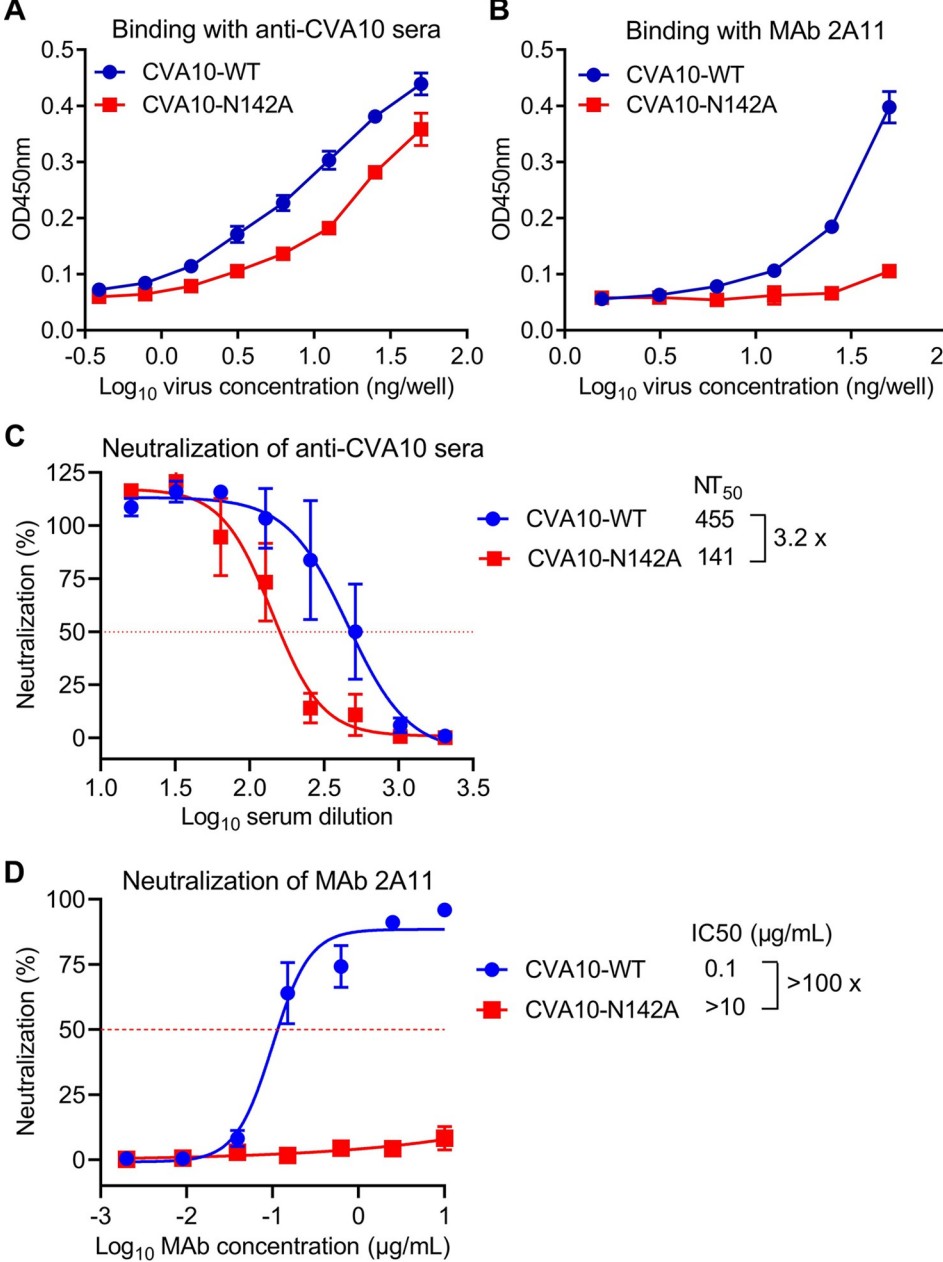

**Fig 7. CVA10-N142A is more resistant to anti-CVA10 sera and MAb 2A11 as compared to CVA10-WT.** (**A-B**) The reactivities of anti-CVA10 sera (**A**) and MAb 2A11 (**B**) to CVA10-WT and CVA10-N142A were determined by ELISA. Purified viral particles were serially diluted and coated onto ELISA plates. Data are mean ± SD of triplicate wells. (**C-D**) Neutralization activities of anti-CVA10 sera (**C**) and MAb 2A11 (**D**) against CVA10-WT and CVA10-N142A were determined. CVA10 viruses were incubated with two-fold serial dilutions of anti-CVA10 antibodies for 1 h before adding to RD cells. Cell viability was measured 3 days after culture. The red dashed line indicates $NT_{50}$ or $IC_{50}$. Data are mean ± SEM of four replicate wells.

viral virulence in vivo. Moreover, N2142 is a key residue for the epitopes recognized by anti-CVA10 neutralizing antibodies.

The notion that N2142 of CVA10 is a key receptor-binding residue is strongly supported by experimental evidences. The evidences are as follows: (1) the naturally occurring double

mutant N2142D+N3183Y was highly resistant to neutralization by KRM1-Fc as compared with CVA10-WT and the single mutant N3183Y (Fig 2A); (2) the mutant N2142A was highly resistant to KRM1-Fc compared to CVA10-WT (Fig 2E); (3) The CVA10-N2142A mutant showed significantly reduced binding to human and mouse KRM1 proteins compared to CVA10-WT (Figs 4F and 6D). The other mutations N3183Y, T1219I, and S1238G were shown not to be responsible for the soluble KRM1 receptor-resistant phenotype (Fig 2), and these mutations are possibly caused by random, spontaneous errors that occur during CVA10 (RNA virus) replication. For the mutant screening experiment, mutants in well #1 and #2 showed complete and partial resistance to neutralization by 100 nM of KRM1-Fc, in line with the fact that the N2142D mutation occurred at a frequency of 9/10 and 1/9 in well #1 and #2, respectively (Fig 1B). In addition, sequence analysis using GenBank database revealed that mutation N2142D was found in only one naturally occurring CVA10 strain (Fig 3D), consistent with the fact that N2142D mutation could decrease the KRM1 receptor binding activity and viral replication fitness (Fig 4). In addition, we investigated the presence of the N2142D mutation in other KRM1-dependent enteroviruses, including CVA2, CVA3, CVA4, CVA5, CVA6, CVA10, and CVA12 [29]. NCBI blast analysis showed that CVA2, CVA4, and CVA6 had a polar amino acid N at position 2142, while CVA3, CVA5, and CVA12 had other polar amino acids (T, S, or Q) at this position. N2142D mutation is not present in all of these naturally occurring virus isolates. This is reasonable because N2142D mutation could substantially impair viral fitness, making it relatively uncommon in nature and extremely difficult to detect.

In this study, we found a naturally occurring mutation N2142D that is detrimental to CVA10 viral adsorption and infection. We do not study whether there are natural mutations that favor CVA10 viral invasion, but it's a very important scientific question. This kind of research could help predict the virus epidemic trend and provide an early warning for emergence of disease epidemics. For example, in the study of SARS-CoV-2 infection, the N501Y substitution in the Alpha variant spike protein was reported to be able to increase the affinity of viral spike protein to cellular receptor and was proven to be a major determinant of the increased transmission of the Alpha variant [31]. Thus, N501Y variation is believed to be a major adaptive spike mutation of concern [31].

Previous studies have reported that KRM1 receptor binds to the canyon of CVA10 [26,28]. Michael G. Rossmann proposed the canyon hypothesis: (1) the canyon is the cellular receptor binding site for enteroviruses; (2) the narrow canyon is inaccessible to antibodies with relatively large size, thus allowing enteroviruses to evade host immune surveillance; (3) residues at the base of the canyon are conserved, allowing for receptor specificity; (4) residues on the rim of the canyon could be mutated, permitting viral evolution and escape [32]. In this study, we found that CVA10 N2142 is a key receptor-binding residue, but it is located at the tip of VP2 EF loop and at the south rim of the canyon and is highly exposed on the virion surface. Although N2142 is surface exposed, it is highly conserved (253/255), revealed by sequence analysis of naturally occurring CVA10 strains in GenBank (Fig 3). Thus, CVA10 N2142 is a counterexample to the canyon hypothesis.

It is reported that KRM1 receptor plays an essential role in CVA10 infection in human cell culture and in mice [29]. Our results showed that the N2142A mutation could impair CVA10 binding to human KRM1 receptor, resulting in significantly reduced viral attachment and infection in RD cells (Fig 4). Moreover, the N2142A mutation could decrease mouse KRM1 receptor binding activity and reduce viral replication in limb muscle and spinal cord of infected mice, leading to lower mortality and less severe clinical symptoms (Figs 5 and 6). These results demonstrate that residue N2142 of CVA10 plays a key role not only in viral binding and infection in cell culture, but also in viral pathogenicity in vivo.

A limitation of our study is that due to time and resource constraints, we only used RD cells for CVA10 research and did not investigate the infectivity of wildtype and mutant CVA10 viruses in other enterovirus-sensitive cell lines. Previous studies report that CVA10 can exhibit quite different growth kinetics when infecting different human cell lines, such as RD (rhabdo-myosarcoma cells), Hela (cervical epithelial cells), HEK 293A (embryonic kidney cells), KB (oral carcinoma cells), SK-N-SH (neuroblastoma cells), and MRC-5 (lung fibroblasts) [33,34]. This phenomenon is probably due to differences in cell surface expression level of the KRM1 receptor. It will be important in the future to examine whether the N2142A mutation would affect CVA10 viral replication in these human cell lines.

Our previous study has demonstrated that VP2 peptide P28, corresponding to residues 136 to 150 of VP2, represents a CVA10-specific linear antigenic site that could induce neutralizing and protective antibody response in mice [35]. In line with the conclusion, N2142, a residue within the P28 peptide region, when mutated decreased viral binding affinity to anti-CVA10 polyclonal antibodies and neutralizing MAb 2A11. Moreover, the N2142A mutation was capable of rendering CVA10 resistant to neutralization by anti-CVA10 sera and MAb 2A11 (Fig 7). These results indicate that N2142 is a critical part of the epitopes recognized by anti-CVA10 neutralizing antibodies (polyclonal and monoclonal antibodies).

Overall, our study reveals that CVA10 residue N2142 plays an important role not only in the interactions with KRM1 receptor and anti-CVA10 neutralizing antibodies, but also in viral virulence in mice. These findings promote a better understanding of the molecular basis of viral attachment and infection and pathogenicity in vivo, and also provide useful information for the development of vaccines and antibody-based treatment against CVA10 infection.

## Materials and methods

### Ethics statement

The mouse studies were approved by the Institutional Animal Care and Use Committee at Shanghai Institute of Immunity and Infection.

### Cells and viruses

Human rhabdomyosarcoma (RD) cells were cultured in DMEM (Gibco, Thermo Fisher Scientific, USA) plus 10% fetal bovine serum (FBS) at 37˚C. CVA10 prototype strain Kowalik (GenBank ID: AY421767) was obtained from ATCC and propagated in RD cells. Viral titers were determined by 50% tissue culture infectious dose ($TCID_{50}$) assay in RD cells.

### Proteins and antibodies

To generate human KRM1 protein, DNA fragment encoding the ectodomain (residues A23 to G373) of human KRM1 was cloned into a modified pcDNA3.4 vector with an N-terminal IL-10 signal sequence and C-terminal human IgG1 Fc and His tag, yielding plasmid pcDNA3.4-hKRM1-Fc. Similarly, to obtain murine KRM1 protein, DNA fragment encoding the ectodomain (residues A23 to G373) of mouse KRM1 was inserted into the modified pcDNA3.4 vector to make plasmid pcDNA3.4-mKrm1-Fc. The plasmids were separately transfected into HEK 293F suspension cells using Polyethylenimine (PEI) Max 40K (Poly-Sciences, USA). Culture supernatants were harvested after 5 days of culture and the His-tagged hKRM1-Fc and mKrm1-Fc fusion proteins were purified using Ni-NTA resin. Protein concentrations were determined by Bradford assay (Bio-Rad).

Polyclonal antibodies were raised against CVA10 by multiple immunization of BALB/c mice with purified CVA10/Kowalik formulated with alum. MAb 2A11 is a mouse IgA

antibody with strong neutralizing activity against CVA10, which was prepared in our lab according to previously described protocols [36,37].

## Generation and sequence analysis of KRM1 neutralization-resistant mutants

Escape mutants were selected according to a previously described protocol [37]. Briefly, 100 $TCID_{50}$ of CVA10/Kowalik was incubated with serially diluted hKRM1-Fc protein for 1 h at 37°C and then added to RD cells grown in 96-well plates. Three days after culture, both the cells and culture supernatants were subjected to 2 cycles of freeze thawing, followed by clarification by centrifugation to remove cell debris. 1 µl of each of the samples (passage 1 [P1] virus) was incubated with 50 nM of hKRM1-Fc and used to infect fresh RD cells. After 3 days of culture, 1 µL of the resulting P2 virus was subjected to another round of selection with 100 nM of hKRM1-Fc, yielding P3 virus. The resultant neutralization escape mutants were plaque purified using low-melting agarose. Afterwards, plaques were picked at random, and viruses from individual plaques were amplified on RD cells. Total cellular RNA was isolated from each culture using TRIzol reagent (Invitrogen, USA), followed by reverse transcription (RT) to synthesize cDNA using M-MLV reverse transcriptase (Promega, USA). The cDNA segment encoding the CVA10 capsid proteins was amplified by PCR and analyzed by DNA sequencing.

## Neutralization assay

Each of the escape mutants was titrated by $TCID_{50}$ assay and then subjected to neutralization assay in 96-well plates. Briefly, 100 $TCID_{50}$/well of wild-type or mutant CVA10 viruses were incubated with 50 µL/well of 2-fold serially diluted hKRM1-Fc protein or anti-CVA10 antibodies for 1 h at 37°C. Next, 100 µL of RD cell suspension (20,000 cells) was added to each well and cultured for 3 days before the examination of cytopathic effect (CPE). Neutralization concentration is defined as the lowest hKRM1-Fc or antibody concentration that could completely inhibit CPE. Immediately after CPE observation, cell viability was determined using CellTiter-Glo 2.0 assay kit (Promega, USA) according to the manufacturer's protocol. Percent neutralization was calculated by the formula: 100 x (relative luminescence unit [RLU] of the sample—RLU of the virus control) / (RLU of the untreated cell control—RLU of the virus control). NT50 or IC50 values were calculated using nonlinear regression curve fit analysis by Graph Pad Prism 8.0.

## Recovery and characterization of wild-type and mutant CVA10 viruses

Full-length genomic cDNA of CVA10/Kowalik was synthesized and cloned in the pVAX1 vector by GenScript (Nanjing, China), yielding plasmid pVAX-CVA10-WT. Single point mutations were introduced into this plasmid using NEBuilder HiFi DNA Assembly Master Mix (NEB) or ClonExpress MultiS One Step Cloning Kit (Vazyme, China), and the desired sequence alterations were confirmed by DNA sequencing. To recover CVA10 viruses, the plasmids containing the wild-type or mutant CVA10 genomic cDNA were co-transfected with the T7 RNA polymerase-encoding plasmid pLVX-T7-RNA-pol into HEK 293T cells using Lipofectamine 2000 (Invitrogen). The CVA10 viruses were collected 3 days post-transfection and propagated once in fresh RD cells. The capsid-coding region of the propagated CVA10 viruses were amplified by RT-PCR and sequenced. The rescued CVA10 viruses were titrated by $TCID_{50}$ assay and then subjected to neutralization assay with hKRM1-Fc protein as described above.

## Structural representation of the interaction between CVA10 and KRM1

The cryo-EM structure of mature CVA10 in complex with KRM1 (PDB: 7BZU) [28] was used for structure analysis in this study. All structural representations were generated using UCSF Chimera software (v1.15) [38]. Interaction surface analysis was performed using PISA server [39].

## Sequence conservation analysis

CVA10 VP1, VP2, and VP3 protein sequences were downloaded from NCBI database (as of April 2023) using BLAST and then aligned using Vector NTI Advance (v11.5.1) and CLC Sequence Viewer software (v8.0).

## Viral growth kinetics

RD cells grown in 24-well plates were infected with CVA10-WT or CVA10-N142A at a multiplicity of infection (MOI) of 0.01 at 37˚C for 1 h. The cells were washed twice with PBS and incubated in fresh DMEM with 1% FBS at 37˚C for 0, 6, 12, 24, 36, or 48 h. Next, both the cells and culture supernatants were subjected to 2 cycles of freeze thawing, and viral titers were determined by $TCID_{50}$ assay.

## Purification and characterization of CVA10-WT and CVA10-N142A

Viral particles were purified according to a previously described protocol [37]. Briefly, RD cells grown in 15-cm dishes were infected with CVA10-WT or CVA10-N142A at a MOI of 0.01. After 3 days of culture, both the cells and culture supernatants were subjected to 2 cycles of freeze thawing, followed by clarification by centrifugation. The resultant supernatants were precipitated and concentrated with 10% polyethylene glycol (PEG) 8000 and 200 mM NaCl at 4˚C overnight. The pellets were collected by centrifugation and resuspended in 0.15 M PBS buffer. The samples were subjected to high-speed centrifugation to remove insoluble materials, and the resulting supernatants were loaded onto 20% sucrose cushion and ultra-centrifuged at 27,000 rpm for 4 h. The resultant pellets were resuspended in 0.15 M PBS buffer and clarified by centrifugation. Next, the clarified supernatants were overlayed on 10–50% sucrose gradients and ultra-centrifuged at 39,000 rpm for 3 h. Following ultracentrifugation, 12 fractions were harvested from top to bottom and analyzed by SDS-PAGE.

To characterize the purified viral particles, the fraction #10 samples were diluted to 50 μg/ml in PBS and loaded onto glow-discharged carbon-coated copper grids and negatively stained with 0.5% uranyl acetate. Images were obtained on a Tecnai G2 Spirit transmission electron microscope (FEI, USA) operated at 200 kV. In addition, the fraction #10 samples were diluted to 1 μg/ml and viral titers were determined by $TCID_{50}$ assay.

## Virus attachment assay

500 μL/well (0.1 μg/mL) of purified CVA10-WT or CVA10-N142A viral particles was added to pre-chilled RD cells in 24-well plates and incubated for 2 h at 4˚C. The cells were washed twice with cold PBS. Next, RNA was isolated from cells by using TRNzol Universal reagent (TIANGEN, China), and cDNA was synthesized using PrimeScript RT reagent Kit (Takara, Japan). Real-time PCR analysis was performed using SYBR Premix Ex Taq kit (Takara) with LightCycler 480 II system (Roche, Switzerland) according to the manufacturers' instructions. CVA10-specific primers were as follows: forward primer, 5′-GAAATGGAGTGTTGGAGGCGA-3′; reverse primer, 5′- TTTCTGTTGTAGTGACGAATG-3′. β-actin primers were as follows: forward primer, 5′-GGACTTCGAGCAAGAGATGG-3′; reverse primer, 5′-AGCACTGTGTT

GGCGTACAG–3′. All results were analyzed using the $2^{-\Delta\Delta Ct}$ method with β-actin as the reference.

## Receptor-binding ELISA

Before ELISA assay, purified human and mouse KRM1-Fc proteins were separately labeled with EZ-Link Sulfo-NHS-LC-LC-Biotin (Thermo Fisher Scientific) and desalted using Zeba spin desalting columns (Thermo Fisher Scientific) according to the manufacturer's instructions.

Purified CVA10-WT or CVA10-N142A viral particles (fraction #10) were serially diluted and coated onto ELISA plates at 4˚C overnight. The plates were blocked with 5% milk in PBS-Tween20 (PBST). Next, biotinylated hKRM1-Fc (50 ng/well), mKrm1-Fc (100 ng/well), or ACE2-Fc [40] (control) proteins were added and incubated at 37˚C for 1 h. After washing, the plates were incubated with horseradish peroxidase (HRP)-conjugated streptavidin (diluted 1:5,000; Proteintech) at 37˚C for 1 h. After color development, absorbance was determined at 450 nm.

## In vivo infection assay

Groups of 2-day-old ICR mice were infected intraperitoneally (i.p.) with 300 $TCID_{50}$/mouse of CVA10-WT or CVA10-N142A. The infected mice were observed daily for survival and clinical score for 14 days. Clinical scores were graded as follows: 0, healthy; 1, reduced mobility; 2, limb weakness; 3, paralysis; 4, death.

To determine viral loads in organs, groups of 2-day-old ICR mice were infected i.p. with 300 $TCID_{50}$/mouse of CVA10-WT or CVA10-N142A. The mice were monitored daily for clinical score and sacrificed 4 days post infection, and limb muscle, spinal cord, and brain were harvested from each mouse, weighed, and homogenized in 400 μL of DMEM with 1% FBS. The samples were clarified by high-speed centrifugation and the resulting supernatants were used for determination of virus titers by $TCID_{50}$ assay.

For pathological analysis, groups of 2-day-old ICR mice were injected with DMEM medium (control) or 300 $TCID_{50}$/mouse of CVA10-WT or CVA10-N142A. The infected mice were observed daily for clinical score and sacrificed at 4 dpi, and limb muscles were collected from each mouse and fixed in 4% paraformaldehyde at room temperature for 24 h. Next, the samples were subjected to hematoxylin and eosin (H&E) staining procedure (Servicebio, China). The samples were visualized under a light microscope, and images were taken for analysis.

## Antibody-binding ELISA

ELISA plates were coated with serially diluted purified CVA10-WT or CVA10-N142A viral particles (fraction #10) at 4˚C overnight and then blocked with 5% milk in PBST. Next, the plates were incubated with 50 μL/well of anti-CVA10/Kowalik sera (diluted 1:10,000) or MAb 2A11 (50 ng/well) and incubated at 37˚C for 2 h. After washing, the plates were incubated with HRP-conjugated anti-mouse IgG (Sigma) or HRP-conjugated anti-mouse kappa (Southern Biotech) and incubated for 1 h at 37˚C. After color development, absorbance was read at 450 nm.

## Acknowledgments

We thank Jinkai Zang and Jiaming Lan (Chinese Academy of Sciences) for critical reading of the manuscript. We thank Rainin, a METTLER TOLEDO Company, for technical support.

## Author Contributions

**Conceptualization:** Chao Zhang.

**Data curation:** Xue Li.

**Formal analysis:** Xue Li, Yuan Tian, Pei Hao, Chao Zhang.

**Funding acquisition:** Chao Zhang.

**Investigation:** Xue Li, Zeyu Liu, Xingyu Yan, Yuan Tian, Kexin Liu, Yue Zhao, Jiang Shao.

**Methodology:** Xue Li, Yuan Tian, Chao Zhang.

**Project administration:** Chao Zhang.

**Resources:** Chao Zhang.

**Software:** Chao Zhang.

**Supervision:** Pei Hao, Chao Zhang.

**Validation:** Chao Zhang.

**Visualization:** Chao Zhang.

**Writing – original draft:** Chao Zhang.

**Writing – review & editing:** Chao Zhang.

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
