## [Decision Letter · Decision Letter 0]

31 Jul 2023

Dear Dr Zhang,

Thank you very much for submitting your manuscript "VP2 residue N142 of coxsackievirus A10 is critical for the interaction with KREMEN1 receptor and neutralizing antibodies and the pathogenicity in mice" for consideration at PLOS Pathogens. As with all papers reviewed by the journal, your manuscript was reviewed by members of the editorial board and by several independent reviewers. The reviewers appreciated the attention to an important topic. Based on the reviews, we are likely to accept this manuscript for publication, providing that you modify the manuscript according to the review recommendations.

Please address all reviewer comments. The Major comment from Reviewer 1 can be addressed with thoughtful discussion added to the text. Regarding major concern #3 from Reviewer 2, addition of a mock-infected control group to the survival curve will not be necessary for the revision.

Sincerely,

Jacob S. Yount

Academic Editor

PLOS Pathogens

Ana Fernandez-Sesma

Section Editor

PLOS Pathogens

Kasturi Haldar

Editor-in-Chief

PLOS Pathogens

orcid.org/0000-0001-5065-158X

Michael Malim

Editor-in-Chief

PLOS Pathogens

orcid.org/0000-0002-7699-2064

Please address all reviewer comments. The Major comment from Reviewer 1 can be addressed with thoughtful discussion added to the text. Regarding major concern #3 from Reviewer 2, addition of a mock-infected control group to the survival curve will not be necessary for the revision.

Reviewer Comments (if any, and for reference):

Reviewer's Responses to Questions

**Part I - Summary**

Reviewer #1: Li et al., reported here the VP2 N142 of CAV10 is essential for viral recognition by KRM1 receptor and also correlates with neutralizing antibodies and pathogenicity. The cellular experiments, rescue experiments, and in vivo experiments conducted in mice collectively offer comparatively substantial data support. Generally this paper is interesting, however some major issues have to be addressed before accepted by PLoS Pathogens.

Reviewer #2: The authors found a residue in VP2 N142 of CVA10, which is essential to interact with KREMEN1 receptor and neutralizing antibodies and viral virulence in mice. This article provides a better understanding of the molecular mechanisms of CVA10 infection and immunity. At this stage, I have the following suggestions and comments.

**Part II – Major Issues: Key Experiments Required for Acceptance**

Reviewer #1: A mutation at a receptor-binding interaction site on VP2 resulted in a reduction of virus infectivity, which is considered not a favorable mutation for the virus. The authors also mention in Figure 3D that this mutation is only present in 1 out of 255 sequences in the NCBI database. This largely weakens the significance of this study. I wonder if the authors have investigated the presence of mutations in N2142D in other enteroviruses that rely on KRM1 invasion? Previous studies suggest that the same virus exhibits different titers when infecting different cells due to differences in protein expression levels on different cell surfaces, what is the infectivity of the mutation in N2142 in other enterovirus-sensitive cell lines? Furthermore, is there other mutations in nature that are variations favoring virus invasion, which would likely be a trend for further virus epidemic expansion and would provide an early warning scenario for disease epidemics.

Reviewer #2: Major concern:

1. Line 232-238, the author made a conclusion of similar copy number between CAV10 WT and N142A based on the comparable particle number in one micrograph. It is incorrect, because negative stain electron microscopy is not a quantitative tool.

2. Figure 4C, the particles in left panel seem solid, corresponding to mature CAV10, while those in the right panel look like empty particles belonging non-infection virus. This should be confirmed.

3. Figure 5A, it is hard to make a conclusion of survival assay between CVA10 WT and N142A infection without a control group.

4. The authors harvested tissues from limb muscle, spinal cord, and brain for CVA10 viral load determination. Are there any supportive studies?

5. Line 195-205, to analysis the conservation of key sites of CVA10 VP1 VP2 and VP3, the authors align numbers of sequences. The datasets of VP2 and VP3 are much less than VP1, which is unreasonable.

**Part III – Minor Issues: Editorial and Data Presentation Modifications**

Reviewer #1: 1. Line 75: Species names should be italicized, e.g. Picornaviridae

2. Line176-183 & Fig 3C & line 324: The residue N2142 forms hydrogen bonds with D88 and D90 in KRM1 but not with G89. Moreover, the N2142D mutation results in strong repulsive interactions with D88 and D90 in KRM1.

3. Fig 5C: The negative stained EM photos in your paper are a bit blurry. Please consider replacing them with clearer ones.

Reviewer #2: Minor concern:

1. Line 121-125, the authors claimed to isolate 10 isolates from well #1, while only 9 mutations were described, why is the last one not mentioned?

2. In figure 1 B, the names of mutations, such as VP2 N142D and VP3 N183Y, are different from the text and other figures naming N2142D, and N3183Y, respectively, which is not reader-friendly. It should be better to unify the names.

3. Figure 4E, the units between virus titer and fold change are different, therefore, the table should be modified.

4. Figure 4F, it seems more logical to show the three sample in a better order, such as CVA10-WT, CVA-N142A, and Cell only.

5. Figure 6b is missed in the main text.

PLOS authors have the option to publish the peer review history of their article (what does this mean?). If published, this will include your full peer review and any attached files.

Reviewer #1: No

Reviewer #2: **Yes: **Xiangxi Wang

Figure Files:

Data Requirements:

Reproducibility:

References:

---

## [Editor Report · Decision Letter 1]

5 Sep 2023

Dear Dr Zhang,

We are pleased to inform you that your manuscript 'VP2 residue N142 of coxsackievirus A10 is critical for the interaction with KREMEN1 receptor and neutralizing antibodies and the pathogenicity in mice' has been provisionally accepted for publication in PLOS Pathogens.

Best regards,

Jacob S. Yount

Academic Editor

PLOS Pathogens

Ana Fernandez-Sesma

Section Editor

PLOS Pathogens

Kasturi Haldar

Editor-in-Chief

PLOS Pathogens

orcid.org/0000-0001-5065-158X

Michael Malim

Editor-in-Chief

PLOS Pathogens

orcid.org/0000-0002-7699-2064
---

## [Editor Report · Acceptance letter]

18 Sep 2023

Dear Dr Zhang,

We are delighted to inform you that your manuscript, "VP2 residue N142 of coxsackievirus A10 is critical for the interaction with KREMEN1 receptor and neutralizing antibodies and the pathogenicity in mice," has been formally accepted for publication in PLOS Pathogens.

Best regards,

Kasturi Haldar

Editor-in-Chief

PLOS Pathogens

orcid.org/0000-0001-5065-158X

Michael Malim

Editor-in-Chief

PLOS Pathogens

orcid.org/0000-0002-7699-2064